# Sex-Dependent Differences in the Ischemia/Reperfusion-Induced Expression of AMPA Receptors

**DOI:** 10.3390/ijms25042231

**Published:** 2024-02-13

**Authors:** Lindsay M. Achzet, Darrell A. Jackson

**Affiliations:** Department of Pharmaceutical Sciences and Molecular Medicine, Washington State University—Health Sciences, Spokane, WA 99201, USA; lindsay.achzet@gmail.com

**Keywords:** ischemic/reperfusion injury, AMPA receptor, hippocampus, GluA2, oxygen glucose deprivation, degradation, mRNA

## Abstract

Following ischemia/reperfusion, AMPA receptors (AMPARs) mediate pathologic delayed neuronal death through sustained expression of calcium-permeable AMPARs, leading to excitotoxicity. Preventing the surface removal of GluA2-containing AMPARs may yield new therapeutic targets for the treatment of ischemia/reperfusion. This study utilized acute organotypic hippocampal slices from aged male and female Sprague Dawley rats and subjected them to oxygen-glucose deprivation/reperfusion (OGD/R) to examine the mechanisms underlying the internalization and degradation of GluA2-containing AMPARs. We determined the effect of OGD/R on AMPAR subunits at the protein and mRNA transcript levels utilizing Western blot and RT-qPCR, respectively. Hippocampal slices from male and female rats responded to OGD/R in a paradoxical manner with respect to AMPARs. GluA1 and GluA2 AMPAR subunits were degraded following OGD/R in male rats but were increased in female rats. There was a rapid decrease in GRIA1 (GluA1) and GRIA2 (GluA2) mRNA levels in the male hippocampus following ischemic insult, but this was not observed in females. These data indicate a sex-dependent difference in how AMPARs in the hippocampus respond to ischemic insult, and may help explain, in part, why premenopausal women have a lower incidence/severity of ischemic stroke compared with men of the same age.

## 1. Introduction

Approximately 800,000 people each year in the United States suffer from a stroke [1]. Ischemic stroke, the most prevalent form of stroke, occurs when a vessel in the brain is occluded, resulting in decreased or absent blood flow. The current treatment for ischemic stroke is reperfusion to the infarcted area as quickly as possible, which can be accomplished in some patients by treating with a thrombolytic, such as tissue plasminogen activator (tPA) [2]. While reperfusion is necessary, this act also results in further tissue damage. During ischemia, there is a decrease in ATP availability from the lack of glucose being delivered to the infarcted brain tissue. This lack of ATP results in the disruption of ionic gradients that maintain neuronal homeostasis, resulting in a massive release of neurotransmitters, including glutamate. Excess stimulation of ionotropic glutamate receptors contributes to delayed neuronal death in vulnerable CA1 pyramidal neurons of the hippocampus [3,4,5,6,7,8,9,10,11,12].

AMPA receptors (AMPARs) are glutamatergic ligand-gated cation channels composed of GluA1-4 subunits. The majority of AMPARs in the hippocampus are GluA1—GluA2 or GluA2—GluA3 AMPARs [13]. The presence of the GluA2 subunit renders the AMPAR channel impermeable to divalent cations such as calcium, due to a post-transcriptional modification by the RNA-editing enzyme adenosine deaminase enzyme 2 (ADAR2) [14,15]. An increase in GluA2-lacking, calcium-permeable AMPARs at the plasma membrane plays a role in the induction of long-term potentiation (LTP), or synaptic strengthening [16]. This is typically a short-lived change in AMPAR composition at the plasma membrane, but during ischemia/reperfusion, there is an increase in the amount of GluA2-lacking calcium-permeable AMPARs [11].

This composition switch in AMPARs following ischemia/reperfusion from GluA2-containing AMPARs to GluA2-lacking AMPARs leads to a cytotoxic accumulation of intracellular calcium. This increase in calcium results in excitotoxicity, a process that is responsible for a large proportion of the neuronal death associated with ischemic stroke [17]. Blocking calcium-permeable AMPARs has been shown to decrease cell death in the hippocampus following ischemia/reperfusion [8,11,18].

During long-term depression (LTD), or weakening of synapses, AMPARs are removed from the plasma membrane and degraded by the lysosome [19,20]. In primary hippocampal neuronal cultures, GluA2 AMPAR subunits are degraded by the lysosome following oxygen-glucose deprivation/reperfusion (OGD/R), an in vitro model for ischemia/reperfusion injury [21].

In this study, we used acute organotypic hippocampal slices prepared from aged (10–12 months of age) male and female rats, and exposed these hippocampal slices to OGD/R, our ex vivo model for ischemia/reperfusion. We found a sex-dependent difference in how AMPARs respond to OGD/R in the hippocampus. Males underwent an OGD/R-induced decrease in GluA1 and GluA2 AMPAR subunits, whereas females had an increase in GluA1 and GluA2 protein levels with OGD/R. Our work has uncovered a novel sex-dependent regulation of AMPAR subunits during ischemia/reperfusion. Post-menopausal females are more susceptible to stroke, and the outcomes tend to be worse compared with pre-menopausal females and age-matched males. AMPARs may play a role in this sex- and age-dependent susceptibility to ischemia/reperfusion injury. Understanding the mechanisms modulating the hippocampal responses to ischemia/reperfusion are critical and can lead to the discovery of new treatments, including, potentially, sex-specific treatments to improve patients’ outcomes post-stroke.

## 2. Results

### 2.1. Sex Differences in GluA1 and GluA2 AMPAR Protein Levels with OGD/R

We performed 40 min of OGD on male and female aged rat hippocampal slices, and allowed 0, 30, 60, or 120 min of reperfusion. The normoxic hippocampal slices were time-matched to the 120-min reperfusion time-point. In male hippocampal slices, GluA1 AMPAR subunit protein levels decreased with OGD/R treatment, maximally at the 120-min endpoint (Figure 1A,B). This contrasts with female hippocampal slices where GluA1 AMPAR protein levels increased with OGD/R treatment, maximally at 120 min of reperfusion (Figure 1C,D).

It is well established that GluA2 AMPAR protein levels are decreased in CA1 hippocampal pyramidal neurons following ischemia/reperfusion [21,22,23]. This is consistent with the results we observed in male hippocampal slices with OGD/R treatment. There was a significant decrease in GluA2 AMPAR protein levels with OGD/R treatment in male aged rat hippocampal slices at 120 min of reperfusion. However, in female animals, there was a significant increase in GluA2 AMPAR protein levels with OGD/R treatment at 120 min of reperfusion (Figure 1).

Interestingly, GluA3 levels remained unchanged in both male and female hippocampal slices with OGD/R treatment (Figure 2). This contrasts with previous reports indicating GluA3 is also degraded in CA1 hippocampal pyramidal neurons following OGD/R [21]. Since GluA3 protein levels did not change with OGD/R treatment, we only examined GluA1 and GluA2 for the remainder of the experiments. GluA4 protein levels were not examined in this study.

### 2.2. Sex Differences in the OGD/R-Induced Internalization of GluA1 and GluA2 AMPAR Subunits

It has been previously reported that GluA1 and GluA2 internalize with OGD/R in young male rat hippocampal slices [22]. We sought to determine whether GluA1 and GluA2 surface levels change with OGD/R exposure in an aged rat model system. Since both GluA1 and GluA2 were degraded with OGD/R in male hippocampal slices, we sought to determine whether these proteins were endocytosed prior to degradation. Utilizing a common technique, biotinylation, the amount of GluA1 and GluA2 AMPAR subunits at the cell membrane surface was measured. Surface GluA1 levels in male hippocampal slices decreased with OGD/R treatment, significantly at 60 and 120 min of reperfusion (Figure 3). In male hippocampal slices, there was an initial increase in GluA2 with OGD treatment and then there was significantly less GluA2 at the membrane with 120 min of reperfusion (Figure 4). GluA1 surface protein levels did not change with OGD/R in female hippocampal slices (Figure 5). However, the amount of membrane GluA2 trended towards significantly decreasing with OGD/R in female hippocampal slices (Figure 4
*p* = 0.0559).

### 2.3. The OGD/R-Induced Increase in GluA1 and GluA2 in Females Occurs from Newly Synthesized AMPA Receptors

Utilizing female hippocampal slices, we pre-treated with cycloheximide to determine whether inhibition of protein synthesis would block the OGD/R-induced increase in GluA1 and GluA2 AMPAR subunits. Indeed, the OGD/R-induced increase in both GluA1 and GluA2 AMPAR subunits was abolished with cycloheximide treatment (Figure 5 and Figure 6). However, this did not lead to a decrease in AMPAR subunits following OGD/R, indicating that there may be a sex-dependent difference in AMPAR trafficking with OGD/R in the hippocampus.

### 2.4. Sex Differences in the mRNA Expression of AMPA Receptors with OGD/R

Lastly, we sought to examine whether the expression of AMPAR subunits was altered with OGD/R treatment in male and female hippocampal slices. Utilizing RT-qPCR, we measured the expression of *GRIA1* (GluA1), *GRIA2* (GluA2), *GRIA3* (GluA3), *GRIA2Q/R* (edited-GluA2), and *ADARB2* (ADAR2) in male and female hippocampal slices following OGD/R treatment. Consistent with our previous work examining protein levels in this study, the levels of *GRIA1* and *GRIA2* were decreased in male hippocampal slices with OGD/R, and *GRIA3* levels did not change (Figure 7).

Unexpectedly, the expression levels of *GRIA1* and *GRIA2* were unchanged with OGD/R treatment in female hippocampal slices (Figure 7). Taken into consideration with the previous experiment (Figure 5 and Figure 6), this may indicate that transcription and translation are increased with OGD/R treatment in female hippocampal slices to result in a total increase in GluA1 and GluA2 protein levels with reperfusion. Previous reports indicate that there is a decrease in *ADARB2* with OGD/R treatment, which we confirmed with our results in male hippocampal slices, but not in female hippocampal slices. Accordingly, the amount of edited *GRIA2* also decreased with OGD/R treatment in male hippocampal slices, but not female hippocampal slices (Figure 8).

## 3. Discussion

We have previously shown that GluA2-containing AMPARs undergo internalization and degradation following OGD/R exposure to hippocampal slices obtained from 8-week-old male rats [22,23,24]. However, to date, there has been limited, if any, investigation on the effects of OGD/R on hippocampal GluA2-containing AMPAR trafficking in middle-aged (10–12-month-old) or aged (>18-month-old) animal subjects. In addition to the lack of age-related ischemic/reperfusion GluA2-containing AMPAR trafficking investigations, OGD/R effect on female GluA2-containing AMPAR trafficking and expression is poorly understood. Therefore, we investigated whether there exists sexual dimorphism in the expression and trafficking of GluA2-containing AMPARs following OGD/R. Our results from this study have identified sex-dependent differences in the way ischemia/reperfusion affects hippocampal AMPARs at both the transcript level and the protein level. Perhaps most interesting is the opposite nature of how GluA1 and GluA2 AMPAR subunits respond to OGD/R in male versus female aged rat hippocampal slices. In males, GluA1 and GluA2 AMPAR subunits decrease with OGD/R, maximally at 120 min of reperfusion. However, in females, GluA1 and GluA2 AMPAR subunits increase with OGD/R treatment, maximally at 120 min of reperfusion. We have previously demonstrated that GluA2, but not GluA1, decreases with OGD/R in 8-week-old male hippocampal slices [22,23,24]. Thus, there also appears to be an age-dependent difference in how GluA1 AMPAR subunits respond to the stress of ischemia/reperfusion.

It has been well established that during LTP there is a temporary increase in the amount of calcium-permeable AMPARs [16], but more recently it has been demonstrated that hippocampal long-term depression also involves a transient increase in the amount of synaptic calcium-permeable AMPARs [25]. 

Previously, we and others have shown that GluA1 and GluA2 AMPAR subunits undergo OGD/R-induced internalization in hippocampal slices [22,23,24] or primary hippocampal neuronal cultures [21]. OGD/R-induced internalization of the GluA2 subunit led to the degradation of this subunit. In contrast, ODG/R-induced internalization of GluA1 AMPAR subunit did not lead to degradation of this subunit in 8-week-old hippocampal male slices [22]. In this study, we observed internalization of GluA1 with OGD/R, maximally at 120 min in males, but in females, surface GluA1 levels did not change. Interestingly, we observed an increase in the amount of GluA2 AMPAR subunits with 40 min of OGD in male hippocampal slices, prior to surface removal of GluA2-AMPAR subunits, maximally at 120 min of reperfusion. In females, we found a slight decrease in GluA2 surface levels, trending towards significance. This is contradictory to what we observed in male hippocampal slices. The increase in surface levels of GluA2 could be due to homomeric GluA2 AMPARs or GluA2/3 AMPARs in a typical LTP mechanism. However, we did not observe a transient increase in the amount of GluA1 surface levels as was expected. This may be due to the slower recycling of GluA1 homomeric or GluA1/3 AMPARs that occurs during basal conditions compared with GluA2/GluA3 AMPARs [26,27]. It is also possible that the experiments may have missed a critical time-point during OGD where there could be a transient increase in the amount of GluA1 AMPAR subunits; GluA1-containing (GluA2-lacking) AMPARs are rapidly delivered to the synapse and then quickly internalized and substituted with GluA2-containing AMPARs [28]. This could explain why there was a transient increase in surface GluA2 AMPAR subunits in males, but not GluA1 AMPAR subunits with OGD treatment.

Although there was not an increase in surface expression of GluA1 and GluA2 AMPARs subunits in female hippocampal slices subjected to OGD/R, there was an increase in total protein levels of GluA1 and GluA2 AMPAR subunits. OGD/R-induced increase in GluA1 and GluA2 in females’ hippocampal slices was blocked with pretreatment of cycloheximide, indicating that de novo protein synthesis of these subunits may have been occurring. This increase in AMPARs could be due to an increase in local translation, as neurons have the machinery required for protein synthesis in dendrites [29]. Whether this increase in GluA1 and GluA2 protein expression occurs at local neuronal synaptic sites in the female hippocampus requires further investigation.

Adenosine deaminase acting on RNA (ADAR2), a nuclear enzyme responsible for the post-transcriptional editing of GluA2 (Q/R), protein and mRNA levels in the hippocampus are significantly reduced following ischemic/reperfusion injury [30]. We examined the mRNA expression levels of AMPAR subunits GluA1, GluA2, GluA3, and edited GluA2Q/R, as well as the RNA-editing enzyme ADAR2 (*ADARB2*). There was a rapid OGD/R-induced decrease in GluA1, GluA2, edited GluA2Q/R, and *ADARB2* expression in males, but not in females. This further confirms that the OGD/R-induced increase in protein levels of GluA1 and GluA2 in females may be due to an increase in protein translation, and not necessarily an increase in mRNA expression. The ischemia/reperfusion-induced decrease in the expression of AMPAR subunits has been observed in other studies [14,31]. This decrease in ADAR2 mRNA levels in male hippocampal slices may underlie the susceptibility of hippocampal neuronal cells to calcium-permeable AMPAR-mediated excitotoxicity. In contrast, the lack of changes in GluA2 (Q/R) and ADAR2 mRNA levels in female hippocampal slices subjected to OGD/R indicate a lesser role of calcium-permeable AMPARs in mediating excitotoxicity. Both the transcriptional and translational dysregulation of AMPAR subunits appear to have key roles in the trafficking of AMPARs with ischemia/reperfusion.

In summary, this study found an OGD/R-induced increase in GluA1 and GluA2 in females, which was not observed in males. This sex-dependent difference is interesting and warrants further investigation. Ultimately, these data help elucidate the degradation pathway of AMPARs with ischemia/reperfusion in males and uncovered a sex difference in AMPAR trafficking. These results may help explain, in part, why premenopausal females are more protected from ischemic stroke in their lifespan compared to age-matched males, and the precise regulation of AMPARs, with respect to sex, following ischemia stroke is an exciting new area for future investigation.

## 4. Materials and Methods

### 4.1. Animals

Male and female Sprague Dawley retired breeder rats (10–12 months of age; Envigo, Indianapolis, IN, USA) were used in this study. This age was chosen to best reflect humans of approximately 45 years old. Animals were allowed free access to food and water for the duration of this study. All animals were maintained according to the National Institutes of Health Guide for the Care and Use of Laboratory Animals. All animal studies were approved by the Washington State University Institutional Animal Care and Use Committee (IACUC).

### 4.2. Reagents

All antibodies used in this study are listed in Table 1 below. Cycloheximide was purchased from Acros Organics (Carlsbad, CA, USA).

### 4.3. Preparation of Acute Rat Hippocampal Slices

The rats were anesthetized with isoflurane and decapitated. The brain was removed and placed in ice-cold Hank’s Buffered Salt Solution (HBSS; Gibco, Amarillo, TX, USA) for no more than 30 s. Both hippocampi were rapidly dissected on ice and 350 µm-thick coronal slices were prepared using a McIlwaine tissue chopper (Stoelting Co., Wood Dale, IL, USA). Slices were equilibrated in oxygenated (95% O_2_, 5% CO_2_) artificial cerebrospinal fluid (aCSF; 124 mM NaCl, 2.5 mM KCl, 26 mM NaHCO_3_, 1.25 mM NaH_2_PO_4_, 2.5 mM CaCl_2_, 1.5 mM MgCl_2_, 10 mM D-glucose, pH 7.4) at 37 °C for 60 min prior to oxygen-glucose deprivation/reperfusion (OGD/R). Fresh aCSF was replaced every 15 min during the equilibration period and every 30 min during the reperfusion period.

### 4.4. Oxygen-Glucose Deprivation/Reperfusion of Hippocampal Slices

Following equilibration, hippocampal slices were placed in deoxygenated, glucose-free aCSF (aCSF with 10 mM sucrose and no glucose, pH 7.4) and incubated for 40 min in a hypoxic glove box (Coy Laboratories, Grass Lake, MI, USA) containing 100% N_2_. The glucose-free aCSF solution used for OGD was de-oxygenated using argon gas under vacuum to ensure completely anoxic aCSF. Following OGD, slices were transferred back to oxygenated glucose-containing aCSF for the time-periods indicated for each experiment (0, 30, 60, or 120 min). Normoxic controls remained in glucose-containing aCSF throughout the entirety of each experiment and time-matched to the last reperfusion time point of OGD/R-subjected slices (Figure 9).

### 4.5. Lysate Preparation

Upon completion of OGD/R or normoxia, the hippocampal slices were lysed (50 mM Tris, 140 mM NaCl, 5 mM EDTA, 1% triton x-100, 1% Halt protease, and phosphatase inhibitor cocktail (ThermoFisher Scientific, Asheville, NC, USA) and 1 mM phenylmethylsulfonyl fluoride) by Dounce homogenization followed by sonication for 3 separate 5 s bursts at 25% power output (VirTis Ultrasonic Cell Disrupter; Gardiner, NY, USA). Samples were then centrifuged at 13,000× *g* for 10 min (4 °C). The supernatant was collected and a bicinchoninic acid assay (BCA) was performed to determine protein concentration. Samples were denatured in NuPAGE LDS (Fisher Scientific, Waltham, MA, USA) with heat (100 °C) for 10 min and resolved with sodium dodecyl sulfate polyacrylamide gel electrophoresis (SDS-PAGE). Samples were transferred to a nitrocellulose membrane (Bio-Rad, Berkeley, CA, USA) for subsequent detection by immunoblotting.

### 4.6. Immunoblotting

Blots were blocked for 1 h at room temperature with either SuperBlock (Thermo Fisher Scientific, Asheville, NC, USA) for phospho-antibody detection or 5% non-fat dry milk in Tris-buffered saline, 0.1% Tween 20, pH 7.5 (TBS-T) for non-phosphorylated antibodies. After blocking, blots were incubated with primary antibody overnight at 4 °C at the concentration indicated in Table 1. Immunoreactive bands were visualized and captured utilizing a FujiFilm imaging system (Edison, NJ, USA) using enhanced chemiluminescence after addition of HRP-conjugated secondary antibodies (Rabbit-HRP or Mouse-HRP, Cell Signaling, Danvers, MA, USA). Bands were analyzed via densitometry using Fuji Image-Gauge software (Image Guage V4.22). The relative levels of protein expression were obtained by quantitatively comparing the band intensity (arbitrary units) of the protein of interest with the band intensity (arbitrary units) of the control protein. Additionally, to examine the effect of OGD/R on the proteins of interest, samples were normalized to the normoxic control group. Blots were stripped and reprobed up to 2 times with Restore Plus Western Blotting Stripping Buffer (ThermoFisher Scientific, Asheville, NC, USA). 

### 4.7. Biotinylation of Hippocampal Slices

Acute aged rat hippocampal slices were prepared and treated to OGD/R or time-matched normoxia as previously described in this study. At the indicated time-point, slices were washed with ice-cold Buffer A (25 mM HEPES, 119 mM NaCl, 5 mM KCl, 2 mM CaCl_2_, 2 mM MgCl_2_, 30 mM glucose, pH 7.4) and incubated at 4 °C in Buffer A containing 0.5 mg/mL of sulfo-NHS-SS-biotin (ThermoScientific, Waltham, MA, USA) for 30 min with gentle agitation. After biotin labeling, slices were washed three times in ice-cold Buffer A. Slices were then lysed via sonication with 3 separate 5 s bursts at 25% power output in Buffer A containing 1% protease and phosphatase inhibitor cocktail. The lysate was centrifuged at 100,000× *g* in an ultracentrifuge. The biotin-labeled membrane fraction was then solubilized on ice for 30 min using Solubilization Buffer (10 mM Tris, 150 mM NaCl, 1% protease and phosphatase inhibitor cocktail, 1% Triton x-100, pH 8.0). The biotin-labeled membrane fraction was then sonicated with 3 separate 5 s bursts at 25% power output and a BCA assay was performed to determine protein content. Biotin-labeled membrane proteins (500 µg lysate/500 µL of Solubilization buffer) were then incubated overnight with gentle agitation at 4 °C with 60 µL of magnetic streptavidin beads (New England BioLabs, Ipswich, MA, USA). The streptavidin-biotin bead complex was denatured with Laemmli and heat (80 °C) for 10 min, subjected to SDS-PAGE, and transferred to nitrocellulose membranes for subsequent immunoblotting. Immunoreactive bands were analyzed using Fuji Image-Gauge software.

### 4.8. Reverse Transcription, Quantitative Polymerase Chain Reaction (RT-qPCR)

Acute aged rat hippocampal slices were prepared and treated to OGD/R or time-matched normoxia as previously described above. Total RNA was extracted utilizing RNeasy Plus Universal Mini Kit per manufacturer’s instructions (QIAGEN, Germantown, MD, USA) following the manufacturer’s protocol. Total RNA was reverse transcribed to cDNA with the Invitrogen High-Capacity Reverse Transcriptase Kit according to manufacturer’s protocol (ThermoFisher Scientific, Waltham, MA, USA). Comparative expression of genes of interest was determined by quantitative polymerase chain reaction (qPCR). The conditions for qPCR were as follows: 95 °C for 2 min followed by 40 cycles of 95 °C for 15 s and 60 °C for 60 s. A melting curve was added to the end of the qPCR run to validate the primers. The PCR amplifications of each set of samples were performed in triplicates and averaged. β-Actin (*ActB*) was used as the internal control for normalization of our genes of interest. The relative fold changes in expression of genes of interest under OGD/R conditions were computed in comparison to the normoxic control group using the 2^−ΔΔCt^ algorithm [32]. A one-way ANOVA was performed using the Sidak post hoc test to determine significance. The primers used are in Table 2 below.

### 4.9. Data Analysis

One-way ANOVA with post hoc Sidak test was conducted using GraphPad Prism 8 software (GraphPad Software, San Diego, CA, USA) to determine statistical significance.

## Figures and Tables

**Figure 1 ijms-25-02231-f001:**
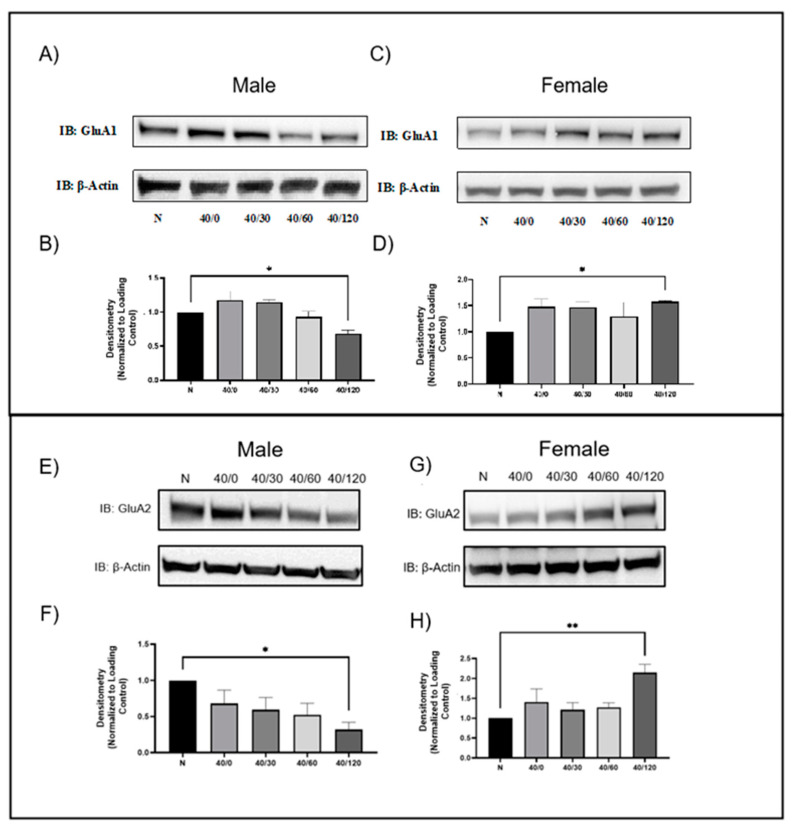
**GluA1 and GluA2 decreased with OGD/R in male hippocampal slices, but GluA1 and GluA2 increased in female hippocampal slices following OGD/R**. (**A**) Representative Western blot demonstrating that GluA1 decreases in male hippocampal slices following OGD/R, maximally at 120 min of reperfusion. (**B**) Quantification of (**A**) of total GluA1 protein levels normalized to β-Actin and normoxic control group (N = 4). (**C**) Representative Western blot demonstrating that GluA1 increases in female hippocampal slices following OGD/R, maximally at 120 min of reperfusion. (**D**) Quantification of (**C**) of total GluA1 protein levels normalized to β-Actin and normoxic control group (N = 4). (**E**) Representative Western blot demonstrating that GluA2 decreases in male hippocampal slices following OGD/R, maximally at 120 min of reperfusion. (**F**) Quantification of (**E**) of total GluA2 protein levels normalized to β-Actin and normoxic control group (N = 4). (**G**) Representative Western blot demonstrating that GluA2 increases in female hippocampal slices following OGD/R, maximally at 120 min of reperfusion. (**H**) Quantification of (**G**) of total GluA2 protein levels normalized to β-Actin and normoxic control group (N = 4). * *p* < 0.05; ** *p* < 0.01; ANOVA with Sidak post hoc test comparing OGD/R conditions corresponding to control (normoxic). Data are expressed as mean ± SEM.

**Figure 2 ijms-25-02231-f002:**
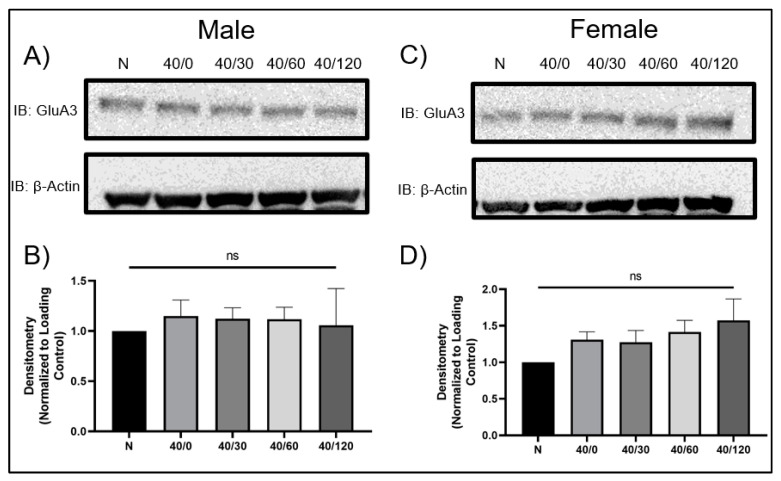
**OGD/R induces degradation of GluA2 in male hippocampal slices and increases GluA2 protein levels in female hippocampal slices**. (**A**) Representative Western blot demonstrating that GluA2 decreases in male hippocampal slices following OGD/R, maximally at 120 min of reperfusion. (**B**) Quantification of (**A**) of total GluA2 protein levels normalized to β-Actin and normoxic control group (N = 4). (**C**) Representative Western blot demonstrating that GluA2 increases in female hippocampal slices following OGD/R, maximally at 120 min of reperfusion. (**D**) Quantification of (**C**) of total GluA2 protein levels normalized to β-Actin and normoxic control group (N = 4). ANOVA with Sidak post hoc test comparing OGD/R conditions corresponding to control (normoxic). Data are expressed as mean ± SEM.

**Figure 3 ijms-25-02231-f003:**
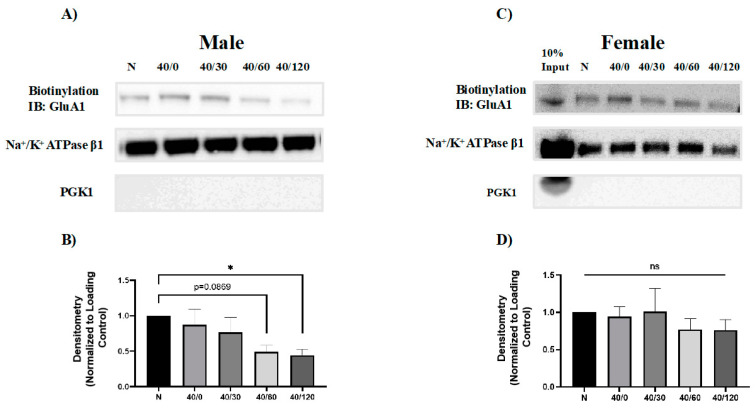
**OGD/R reduces surface levels of GluA1 AMPAR subunits in male hippocampal slices but does not alter surface levels of GluA1 in female hippocampal slices**. Surface proteins were biotinylated and resolved by immunoblotting to examine the amount of GluA1 AMPAR protein at the membrane surface. (**A**) Representative Western blot demonstrating that surface GluA1 decreases in male hippocampal slices with OGD/R. (**B**) Quantification of (**A**) of surface GluA1 protein levels normalized to Na/K ATPase β1 and normoxic control group (N = 4). (**C**) Representative Western blot demonstrating that surface GluA1 does not change in female hippocampal slices following OGD/R. (**D**) Quantification of (**C**) of surface GluA1 protein levels normalized to Na/K ATPase β1 and normoxic control group (N = 4). * *p* < 0.05; ANOVA with Sidak post hoc test comparing OGD/R conditions corresponding to control (normoxic). Data are expressed as mean ± SEM.

**Figure 4 ijms-25-02231-f004:**
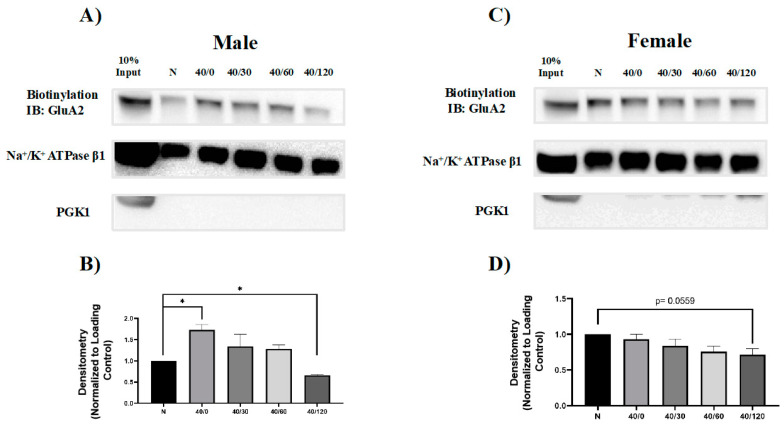
**Sex-dependent differences in amount of surface GluA2 with OGD/R treatment in male and female hippocampal slices.** Surface proteins were biotinylated and resolved by immunoblotting to examine the amount of GluA2 AMPAR protein at the membrane surface. (**A**) Representative Western blot demonstrating an initial increase in surface GluA2 levels with OGD, and then a significant decrease in surface GluA2 with reperfusion at 120 min in male hippocampal slices. (**B**) Quantification of (**A**) of surface GluA2 protein levels normalized to Na/K ATPase β1 and normoxic control group (N = 4). (**C**) Representative Western blot demonstrating that surface GluA2 trends toward decreasing in female hippocampal slices following OGD/R. (**D**) Quantification of (**C**) of surface GluA2 protein levels normalized to Na/K ATPase β1 and normoxic control group (N = 4). * *p* < 0.05; ANOVA with Sidak post hoc test comparing OGD/R conditions corresponding to control (normoxic). Data are expressed as mean ± SEM.

**Figure 5 ijms-25-02231-f005:**
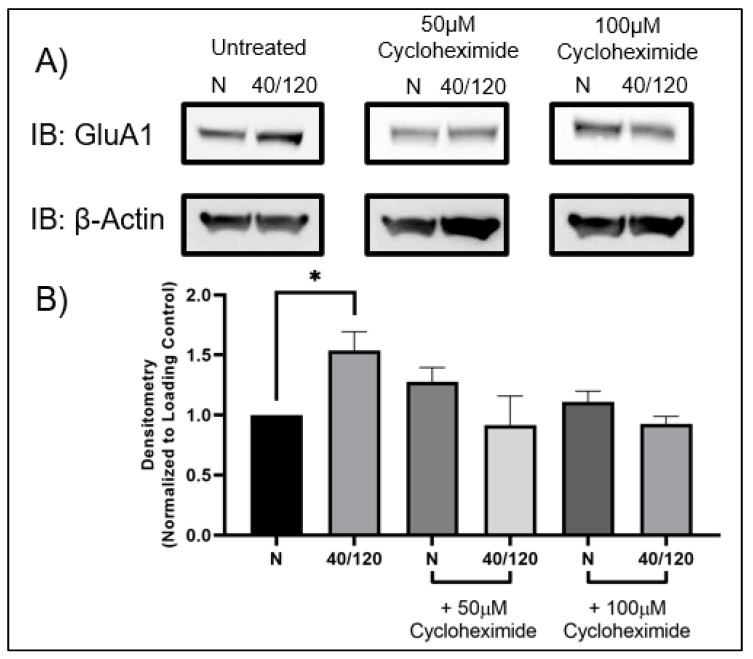
**OGD/R-induced increase in GluA1 is due to increased protein translation in female hippocampal slices.** Female hippocampal slices were exposed to 40 min of OGD and then reperfused for 120 min in the presence or absence of cycloheximide [50 µM] or [100 µM], an inhibitor for protein synthesis. (**A**) Representative Western blot demonstrating that pre-treatment with cycloheximide prevents the OGD/R-induced increase in GluA1 AMPAR subunits in female hippocampal slices. (**B**) Quantification of (**A**) of total GluA1 protein levels normalized to β-Actin and normoxic control group (N = 3). * *p* < 0.05; ANOVA with Sidak post hoc test comparing OGD/R and drug treatments corresponding to control (normoxic). Data are expressed as mean ± SEM.

**Figure 6 ijms-25-02231-f006:**
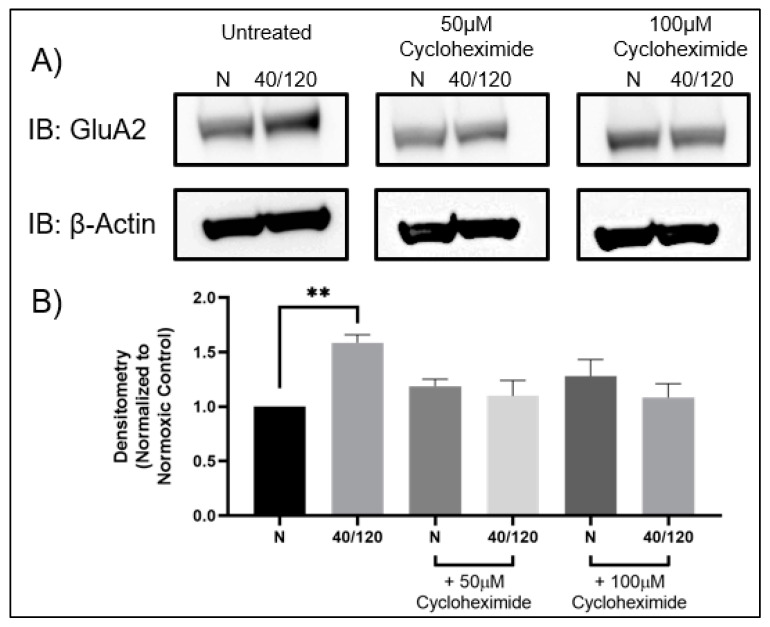
**OGD/R-induced increase in GluA2 is due to increased protein translation in female hippocampal slices.** Female hippocampal slices were exposed to 40 min of OGD and then reperfused for 120 min in the presence or absence of cycloheximide [50 µM] or [100 µM], an inhibitor for protein synthesis. (**A**) Representative Western blot demonstrating that pre-treatment with cycloheximide prevents the OGD/R-induced increase in GluA2 AMPAR subunits in female hippocampal slices. (**B**) Quantification of (**A**) of total GluA2 protein levels normalized to β-Actin and normoxic control group (N = 3). ** *p* < 0.01; ANOVA with Sidak post hoc test comparing OGD/R and drug treatments corresponding to control (normoxic). Data are expressed as mean ± SEM.

**Figure 7 ijms-25-02231-f007:**
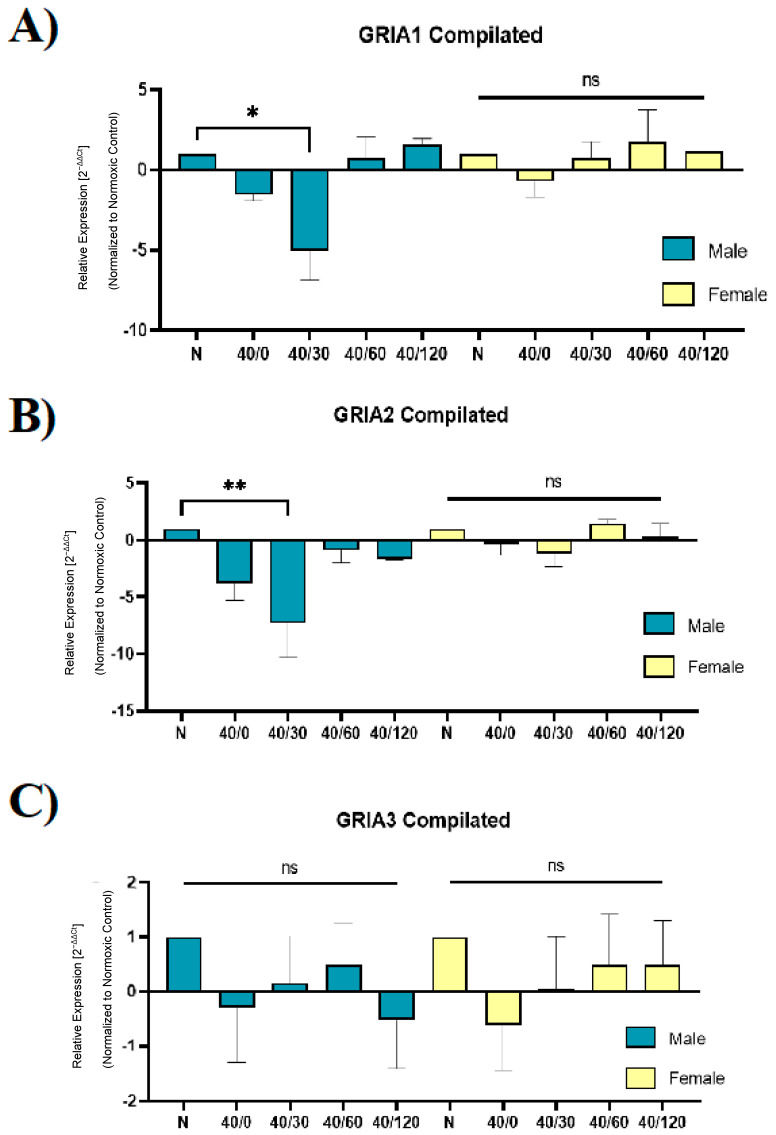
**OGD/R decreases *GRIA1* and *GRIA2* expression following OGD/R in male hippocampal slices, but OGD/R does not significantly affect female *GRIA1* and *GRIA2* expression in hippocampal slices**. Utilizing rt-qPCR, we examined the expression of *GRIA1* (GluA1), *GRIA2* (GluA2), and *GRIA3* (GluA3) with OGD/R treatment. (**A**) Expression of *GRIA1* in male and female hippocampal slices with OGD/R treatment, normalized to normoxic control group and *ActB* (β-Actin) (N = 3). (**B**) Expression of *GRIA2* in male and female hippocampal slices with OGD/R treatment, normalized to normoxic control group and *ActB* (β-Actin) (N = 3). (**C**) Expression of *GRIA3* in male and female hippocampal slices with OGD/R treatment, normalized to normoxic control group and *ActB* (β-Actin) (N = 3). * *p* < 0.05; ** *p* < 0.01; ns denotes no significance. ANOVA with Sidak post hoc test comparing OGD/R values (2^−ΔΔCt^ ) corresponding to normoxic control. Data are expressed as mean ± SEM.

**Figure 8 ijms-25-02231-f008:**
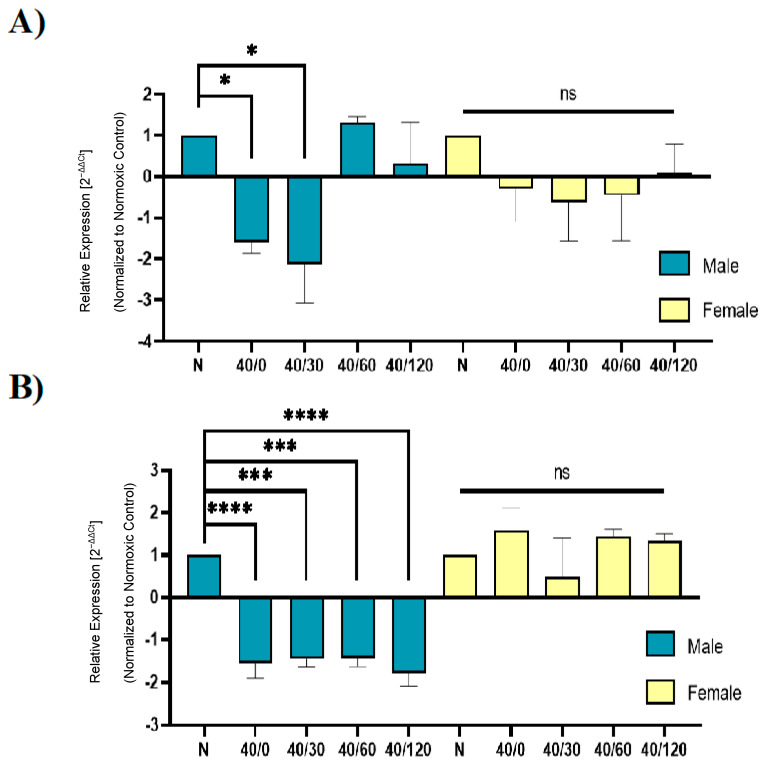
**OGD/R decreases *ADARB2* expression and *GRIA2Q/R* edited (Q/R GluA2) following OGD/R in male hippocampal slices. *ADARB2* and Q/R-edited *GRIA2* expression levels are unaffected with OGD/R in female hippocampal slices**. Utilizing rt-qPCR, we examined the expression of *ADARB2* (ADAR2) and *GRIA2Q/R* (Q/R-edited GluA2) with OGD/R treatment. (**A**) Expression of *ADARB2* in male and female hippocampal slices with OGD/R treatment, normalized to normoxic control group and *ActB* (β-Actin) (N = 3). (**B**) Expression of Q/R-edited *GRIA2* in male and female hippocampal slices with OGD/R treatment, normalized to normoxic control group and *ActB* (β-Actin) (N = 3). * *p* < 0.05; *** *p* < 0.001; **** *p* < 0.0001; ns denotes no significance. ANOVA with Sidak post hoc test comparing OGD/R values (2^−ΔΔCt^ ) corresponding to normoxic control. Data are expressed as mean ± SEM.

**Figure 9 ijms-25-02231-f009:**
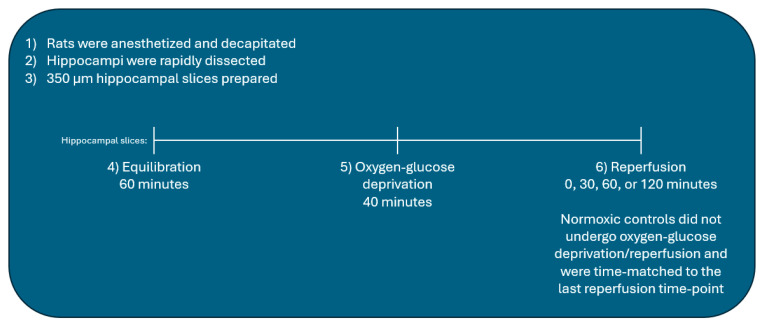
Hippocampal slice experimental methodology.

**Table 1 ijms-25-02231-t001:** Antibodies.

Antibody	Description	Host	Source	Dilution	Product Number
GluA1	AMPA Receptor subunit 1	Rabbit	Cell Signaling	1:1000	13185S
GluA2	AMPA Receptor Subunit 2	Rabbit	Cell Signaling	1:1000	13607S
GluA3	AMPA Receptor Subunit 3	Rabbit	Cell Signaling	1:1000	5117S
β-Actin	Beta-Actin	Rabbit	Cell Signaling	1:1000	8457S
PGK1	Phosphoglycerate kinase 1	Rabbit	Cell Signaling	1:1000	68540S
Na,K-ATPase β1	Na, K-ATPase subunit β1	Rabbit	Cell Signaling	1:1000	44759S

**Table 2 ijms-25-02231-t002:** Primers used in this study [33].

#	Gene	Forward Primer	Reverse Primer	Product bps
1	*Gria1*	GGACAACRCAAGCGTCCAGA	CACAGTAGCCCTCATAGCGG	122
2	*Gria2*	TGGTTTTCCTTGGGTGCCTT	TCGATGGGAGACACCATCCT	170
3	*Gria3*	CCATGCTCTTGTCAGCTTCG	TGTGCTCCTGAACCGTGTTT	178
4	*Gria2 Q/R*	CTACGAGTGGCACACTGAGG	AACCACCACACACCTCCAAC	177
5	*Actb*	GCAGGAGTACGATGAGTCCG	ACGCAGCTCAGTAACAGTCC	74
6	*Adarb2*	GACGACACGCGGGAATATCT	GCCAGCAAGCACCTTCTCTA	131

## Data Availability

All data are present within the manuscript or available by request to corresponding author, Darrell A. Jackson (darrell.jackson@wsu.edu).

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
