# Peer review of "Sex-Dependent Differences in the Ischemia/Reperfusion-Induced Expression of AMPA Receptors"

_ijms, 2024, doi:10.3390/ijms25042231_

Round 1
Reviewer 1 Report
Comments and Suggestions for Authors
The manuscript “Sex-Dependent Differences in the Ischemia/Reperfusion-Induced Expression of AMPA Receptors” by Achzet & Jackson is a research article which examined the effects of oxygen-glucose deprivation/reperfusion (OGD/R) on the expression levels of GluA1 and GluA2 AMPAR subunits in hippocampal slices from male and female rats to clarify the mechanisms underlying the internalization and degradation of GluA2-containing AMPARs. The authors found that GluA1 and GluA2 AMPAR subunits were degraded following OGD/R in male rats but increased in female rats. Furthermore, there was a rapid decrease in GluA1 and GluA2 mRNA levels in the male hippocampus following ischemic insult while this was not observed in females. In general, this article is critical in this field and contains essential findings. The manuscript is well written and scientifically sound. I have several concerns on the paper before publication.
1. In figures 9 and 10, the titles of Y axis are too small to see. Please revise it.
2. In bar graphs, all the data plots should be displayed if possible. The readers can obtain more information from these data.
3. The authors used ANOVA for multiple comparison. Please add F values in the text or figure legends.
4. Page 5, line 18. ”GluA4 protein levels were barely detectable in the hippocampus.” Please describe this issue in the Discussion.
Author Response
Maurizio Battino
International Journal of Molecular Sciences
Editor-in-Chief
Dear Dr. Battino,
I would like to acknowledge the helpfulness of the reviewer’s comments and criticisms of the submitted manuscript (ijms-2776379) to your journal entitled: “Sex-Dependent Differences in the Ischemia/Reperfusion-Induced Expression of AMPA Receptors.” I have followed the suggestions of the reviewers and the manuscript is, therefore, much improved. Each reviewer’s comment is identified and addressed below.

Reviewer 2 Report
Comments and Suggestions for Authors
The paper presents Western immunoblotting data for quantitative assessment of reperfusion in the hippocampus of mice. Many questions arose regarding the results section. Unfortunately, this work cannot be published in its presented form.
Below are the following comments:
1) It is not clear how the statistical assessment was carried out. It is necessary to describe in more detail the procedure for analyzing statistical data. Did the authors use a small group correction? In what units of measurement was optical density assessed? What are the absorbance values for β-actin? How did the authors normalize to β-actin and normoxic controls?
2) In Figure 3, some graininess is visible on the GluA3 gel. What is the reason for the worse quality of the gel for this section of work?
3) It is not clear how the authors calculated the ratio of GluA1 protein levels to GluA2 protein levels. In Fig. 4A and 4B it is absolutely unclear in what units of measurement this ratio is presented and how it was obtained (how the authors measured the optical density of immunoblots). The authors provide very brief explanations in the results, from which it is not clear how the quantitative data were obtained and the algorithm for quantitative evaluation of immunoblots is not clear. Nowhere do the authors provide the values and units of optical density in which they measured and compared the values. Thus, the results of the study are presented very fragmentarily; an algorithm for analyzing data and methods for assessing it is not presented. In the Materials and Methods section, the authors only state that they used one-way ANOVA to evaluate the immunoblotting results. The presented version of the article is not satisfactory from the point of view. presentation of the results and their analysis and cannot be published in its current form.
4) In the Discussion section, the authors repeat the results they have already obtained and discuss them very little with literature data. This approach to discussing your results seems very strange and inappropriate. This section is also poorly written.
Comments on the Quality of English LanguageStylistic editing of the English language is required
Author Response

(The authors gave the same response as above.)

Reviewer 3 Report
Comments and Suggestions for Authors
Dear Editor,
The manuscript entitled “Sex-Dependent Differences in the Ischemia/Reperfusion-Induced Expression of AMPA Receptors”, studied oxygen-glucose deprivation/reperfusion on AMPA receptors GluA1 and GluA2 protein expression by comparing male and female rats hippocampus. The manuscript is well organized and well written. However, there are some issues which need to be considered to support the conclusion and the aim of study.
Major comments:
1. It doesn’t seem that 2 hours of reperfusion could be enough for completion of AMPA receptors protein expression and it is a short time to support translational objectives of the study. Hence, these data just demonstrate sectioned hippocampus AMPAR subunits expression after exposure to OGD in short time frames which is not long enough to conclude as a real pattern of AMPA receptors subunit expression in acute phase of ischemia. In prior studies at least 24 h were applied (DOI: 10.1523/JNEUROSCI.17-16-06179.1997; DOI: 10.1046/j.1471-4159.2003.01986.x; https://doi.org/10.1161/01.STR.0000014205.05597.45.).
2. Also, for having optimized results it would be better to let the tissue adapt to the new conditions inside the culture medium after cutting tissue blood flow through dissection and isolation. Knowing that amino acids are required for protein expression, it intensifies the importance of using culture mediums containing essential amino acids and related supplements needed for protein expression.
3. In figure 4, considering GluA1 is in 40/0 group is about 1.2 of Control (Fig 1) and GluA2 is in 40/0 group is about 0.7 of control (Fig 2), how the authors calculated 6 time increase in GluA1:GluA2 ratio compared to control?
Minor comments:
1. Repeating this intensified structure of definition makes sentences complicated, it needs to change to more simple form: “… expression of calcium permeable AMPARs lacking GluA2 …”, “GluA2-lacking calcium-permeable”, “calcium-permeable, GluA2-lacking AMPARs”.
2. The sentences need to be changed to a simpler form. “Preventing the surface removal and subsequent degradation of GluA2-containing AM-10 PARs…”
3. Graphs resolution is low and hard to read.
4. Similar GluA1 and 2 figures can be merged to make comparison easier for the readers. For example, fig. 1 and 2 can be merged as a one figure, …
5. Instead of repeating this sentence in each figure: “Male and female hippocampal slices were 83 exposed to 40 minutes of OGD and then reperfused for various time-points up to 120 minutes.”, it is suggested to draw a schematic of the study timeline.
6. (N=4)? Does it mean the number of animals in each group? Then, “each group” should be mentioned.

Author Response
Please find responses to all the comments in the attached file.

Round 2
Reviewer 2 Report
Comments and Suggestions for Authors
The authors made the necessary changes to the text of the article in accordance with the critical comments made, and also answered questions that arose during the review of the article. In a revised version, this work can be published in IJMS.
Comments on the Quality of English LanguageEnglish language needs correction
Author Response
Reviewer 2 had not commented for additional changes to the manuscript.
Thank you for your review.
Reviewer 3 Report
Comments and Suggestions for Authors
Author Response
Response to reviewer is uploaded.

Round 3
Reviewer 3 Report
Comments and Suggestions for Authors
Dear Editor,
I appreciate the author's explanation regarding the main concern. Current version of submission (ijms-2776379) meets required criteria for publication.